# Dissection of the host-pathogen interaction in human tuberculosis using a bioengineered 3-dimensional model

Liku B Tezera[1]*, Magdalena K Bielecka[1], Andrew Chancellor[1], Michaela T Reichmann[1], Basim Al Shammari[2], Patience Brace[1], Alex Batty[1], Annie Tocheva[1], Sanjay Jogai[1], Ben G Marshall[1], Marc Tebruegge[1], Suwan N Jayasinghe[3], Salah Mansour[1,4], Paul T Elkington[1,4]*

[1]NIHR Respiratory Biomedical Research Unit, Clinical and Experimental Sciences Academic Unit, Faculty of Medicine, University of Southampton, Southampton, United Kingdom; [2]King Abdullah International Medical Research Center/King Saud bin Abdulaziz University for Health Sciences, Department of Infectious Diseases, MNGHA, Riyadh, Saudi Arabia; [3]BioPhysics Group, UCL Institute of Biomedical Engineering, UCL Centre for Stem Cells and Regenerative Medicine and UCL Department of Mechanical Engineering, University College London, London, United Kingdom; [4]Institute for Life Sciences, University of Southampton, Southampton, United Kingdom

*For correspondence: l.tezera@soton.ac.uk (LBT); p.elkington@soton.ac.uk (PTE)

Competing interests: The authors declare that no competing interests exist.

**Abstract** Cell biology differs between traditional cell culture and 3-dimensional (3-D) systems, and is modulated by the extracellular matrix. Experimentation in 3-D presents challenges, especially with virulent pathogens. *Mycobacterium tuberculosis* (Mtb) kills more humans than any other infection and is characterised by a spatially organised immune response and extracellular matrix remodelling. We developed a 3-D system incorporating virulent mycobacteria, primary human blood mononuclear cells and collagen–alginate matrix to dissect the host-pathogen interaction. Infection in 3-D led to greater cellular survival and permitted longitudinal analysis over 21 days. Key features of human tuberculosis develop, and extracellular matrix integrity favours the host over the pathogen. We optimised multiparameter readouts to study emerging therapeutic interventions: cytokine supplementation, host-directed therapy and immunoaugmentation. Each intervention modulates the host-pathogen interaction, but has both beneficial and harmful effects. This methodology has wide applicability to investigate infectious, inflammatory and neoplastic diseases and develop novel drug regimes and vaccination approaches.

## Introduction

An emerging paradigm in biology is that events in traditional 2-dimensional cell culture often differ from those in 3-dimensional (3-D) culture (*Benam et al., 2015*; *Pampaloni et al., 2007*). Furthermore, the extracellular matrix regulates cell biology (*Yamada and Cukierman, 2007*; *Schwartz and Chen, 2013*; *Parker et al., 2014*) and infection biology differs between 2-D and 3-D systems (*Cheng et al., 2011*; *Barrila et al., 2010*). Human disease occurs in 3-D and in the context of extracellular matrix. Consequently, conclusions drawn from 2-dimensional cell culture systems may not fully reflect events in vivo (*Yamada and Cukierman, 2007*). This presents a challenge to progress from standard culture systems, where cells are grown in 2-D on plastic, to more advanced systems that more faithfully replicate events in man (*Yamada and Cukierman, 2007*). These technical

difficulties are particularly marked in studying infectious diseases, where experiments must have additional levels of containment to prevent the release of pathogens (*Barrila et al., 2010*).

*Mycobacterium tuberculosis* (Mtb) is a pathogen of global significance that continues to kill 1.5 million people per year (*O'Garra et al., 2013*; *Horsburgh et al., 2015*). Unfortunately, despite major investment in research, recent clinical trials and vaccine studies to reduce the global burden of tuberculosis (TB) have been unsuccessful (*Tameris et al., 2013*; *Ndiaye et al., 2015*; *Warner and Mizrahi, 2014*), indicating that the model systems that informed these studies require further refinement. In TB, the host-pathogen interaction is highly complex, with the immune response concurrently necessary for containment of infection but paradoxically also driving immunopathology that leads to lung destruction and transmission (*Russell, 2011*; *Elkington and Friedland, 2015*). The mouse is the principal model system to study TB, but inflammatory conditions in the mouse differ from man (*Seok et al., 2013*), and lung pathology is different in murine Mtb infection (*Young, 2009*). Mtb is an obligate human pathogen and has a very prolonged interaction with host cells, surviving within professional phagocytes (*Russell, 2011*). Therefore, long term human culture experiments are required to investigate pathogenesis. A specific advantage of 3-D cell culture incorporating extracellular matrix is that cellular survival is greatly prolonged (*Buchheit et al., 2012*; *Mueller-Klieser, 1997*). Furthermore, inflammatory signalling in TB granulomas is spatially organised (*Marakalala et al., 2016*), with specific microenvironments (*Mattila et al., 2013*), and the extracellular matrix regulates cell survival in TB (*Al Shammari et al., 2015*), indicating that an optimal system to study human disease will need to be 3-D with extracellular matrix.

We hypothesised that to fully understand the host-pathogen interaction in TB, a 3-D cell culture system that incorporates primary human cells, extracellular matrix, fully virulent Mtb, and multiparameter longitudinal readouts is required. Whilst human cellular models of human granuloma formation have been developed, none have all these characteristics (*Puissegur et al., 2004*; *Lay et al., 2007*; *Kapoor et al., 2013*; *Parasa et al., 2014*). We addressed the technical challenges of performing these experiments at biosafety containment level three by adopting a bioengineering approach (*Workman et al., 2014*). We developed a model system that permits interrogation of the host-pathogen interaction in 3-D in the context of extracellular matrix. We demonstrate that cardinal features of human disease develop and that the host immune response is significantly different when cells are adherent to collagen, favouring the host relative to the pathogen. We investigate emerging therapeutic approaches in the system, and demonstrate that each intervention has both beneficial and likely harmful effects. The model permits the concurrent analysis of multiple outcomes and therefore can be used to develop optimal approaches to address the TB pandemic, and can be applied to diverse infectious, inflammatory and neoplastic diseases.

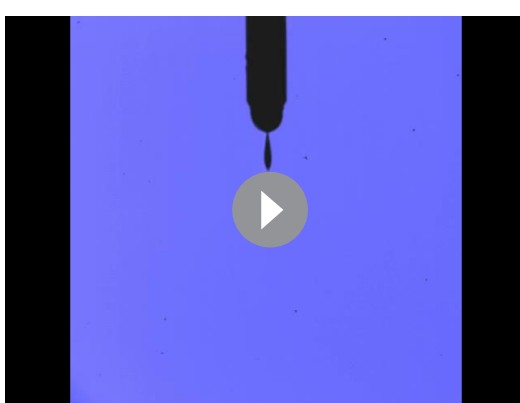

**Video 1.** Generation of microspheres. During the bio-electrospray process, a Phantom v7 high-speed camera, capable of capturing 150000 fps in conjunction with a long-distance microscope lens, was triggered simultaneously with a fibre optic lighting system. Relative video speed 0.15 s.

## Results

### Key features of human tuberculosis develop in the bio-electrospray model

To address the challenges of studying infection of primary human cells with a virulent pathogen within a 3-D extracellular matrix, we optimised the bio-electrospray parameters for stable microsphere generation. PBMCs were isolated from healthy donors, counted and then infected with Mtb that had been cultured in Middlebrook 7H9 broth at a multiplicity of infection of 0.1. After overnight infection, cells were detached, resuspended, and pelleted by centrifugation, and then re-suspended in alginate or alginate-collagen matrix before bioelectrospraying into microspheres using a Nisco Cell Encapsulator (*Video 1*, *Figure 1—figure supplement 1* and *2*). Characterisation of different alginates

indicated that ultrapure medium viscosity guluronate (MVG)-dominant alginate had optimal biophysical properties for electrospraying and minimal immunogenicity (*Figure 1—figure supplement 3*).

Immediately after generation, cells are evenly distributed within microspheres and by day seven cellular aggregates start to form (*Figure 1A and B*). Quantitation demonstrated significantly more aggregates in infected microspheres than unifected microspheres (*Figure 1C*). One day after infection, a quarter of monocytes had phagocytosed Mtb, analyzed by flow cytometric analysis of cells infected with GFP-expressing Mtb (*Figure 1D*). After 14 days of incubation, large cellular aggregates are observed (*Figure 1E*). Survival of Mtb-infected cells in 3-D collagen-alginate microspheres was much greater than in 2-D culture, as analyzed by LDH release (*Figure 1F*) and cellular cytotoxicity assays (*Figure 1G*). An advantage of the model is that cells can be released from the microsphere by decapsulation by divalent cation chelation with EDTA and sodium citrate in HBSS for 10 min, which causes dissolution of the spheres and releases cells for downstream assays. Decapsulation did not significantly affect cell viability, with over 90% cell viability after decapsulation (*Figure 1—figure supplement 4*). Monocytes mature into macrophages in infected microspheres, with greater expression of CD68 (*Figure 1H*). Within the aggregates, multinucleate giant cells typical of human TB develop, stained for CD68 by immunohistochemistry (*Figure 1I*). These multinucleate giant cells are similar to those that occur in human patients with pulmonary TB (*Figure 1J*). T cell differentiation occurs within microspheres, with a progressive increase in the proportion of CD4+ T cells, while the percentage of CD8+ T cells declines (*Figure 1—figure supplement 5*). T cell proliferation does not differ between Mtb-infected and uninfected microspheres.

Next, we measured Mtb growth within microspheres longitudinally by infecting cells with luminescent Mtb expressing the Lux operon, which is genetically modified to constantly luminesce (*Andreu et al., 2010*). Mtb in microspheres without human cells grows relatively slowly, whereas in the presence of PBMCs Mtb proliferates over 24 days, reaching the same luminescence as growth in Middlebrook 7H9 broth (*Figure 2A*). Proteases implicated in TB pathogenesis are upregulated, with MMP-1 gene expression increased within spheres 4 days post infection (*Figure 2B*) and MMP-9 accumulation in media surrounding the spheres peaking at day 7 (*Figure 2C*). This protease activity has a functional effect, causing increased degradation of fluorescently labelled collagen within microspheres (*Figure 2D*). Mtb infection also upregulated gene expression of IFN-$\gamma$ (*Figure 2E*) and drives secretion of multiple pro-inflammatory cytokines, including IL-1$\beta$, IL-12, GM-CSF, IP-10 and MCP-1 analyzed by Luminex multiplex array (*Figure 2F* and *Figure 2—figure supplement 1*), demonstrating that similar cytokines upregulated in human disease are expressed within the microspheres.

## Extracellular matrix integrity regulates the host-pathogen interaction

In patients with TB, the host-pathogen interaction occurs in the context of the collagen-rich lung extracellular matrix (*Elkington et al., 2011*), but most laboratory studies occur in the absence of matrix. The matrix regulates multiple facets of cell biology (*Pampaloni et al., 2007*), and so to determine whether incorporation of matrix into the bio-electrospray system was a critical component of the model, we generated microspheres without collagen or with evenly distributed collagen (*Figure 3A*). Incorporation of type I collagen significantly reduced cell death after Mtb infection, while adherence to elastin increased cell death (*Figure 3B*). We therefore investigated the phenotype in microspheres containing type I collagen further. PBMCs in collagen-containing microspheres had a significantly greater ability to control Mtb proliferation, with a lower Mtb proliferation from day seven in the presence of collagen (*Figure 3C*), further demonstrating that matrix integrity regulates the host-pathogen interaction.

To investigate mechanisms underlying the reduced growth in the presence of collagen, we developed a multiparameter readout. Apoptosis, which is considered a host protective mechanism, was increased in collagen-containing spheres compared to spheres with no collagen (*Figure 3D*). In addition, the NADP-NADPH ratio was higher in collagen-containing spheres (*Figure 3E*), demonstrating divergent cellular energy homeostasis. Secretion of multiple proinflammatory cytokines in microspheres was increased in the presence of collagen, including IL-1$\beta$, TNF-$\alpha$, IFN-$\gamma$, IL-6, IL-8 and MCP-1 (*Figure 3F–K*). Therefore, the host-pathogen interaction is markedly different in the presence of collagen, with improved control of Mtb growth, greater cell survival and altered energy balance and cytokine secretion.

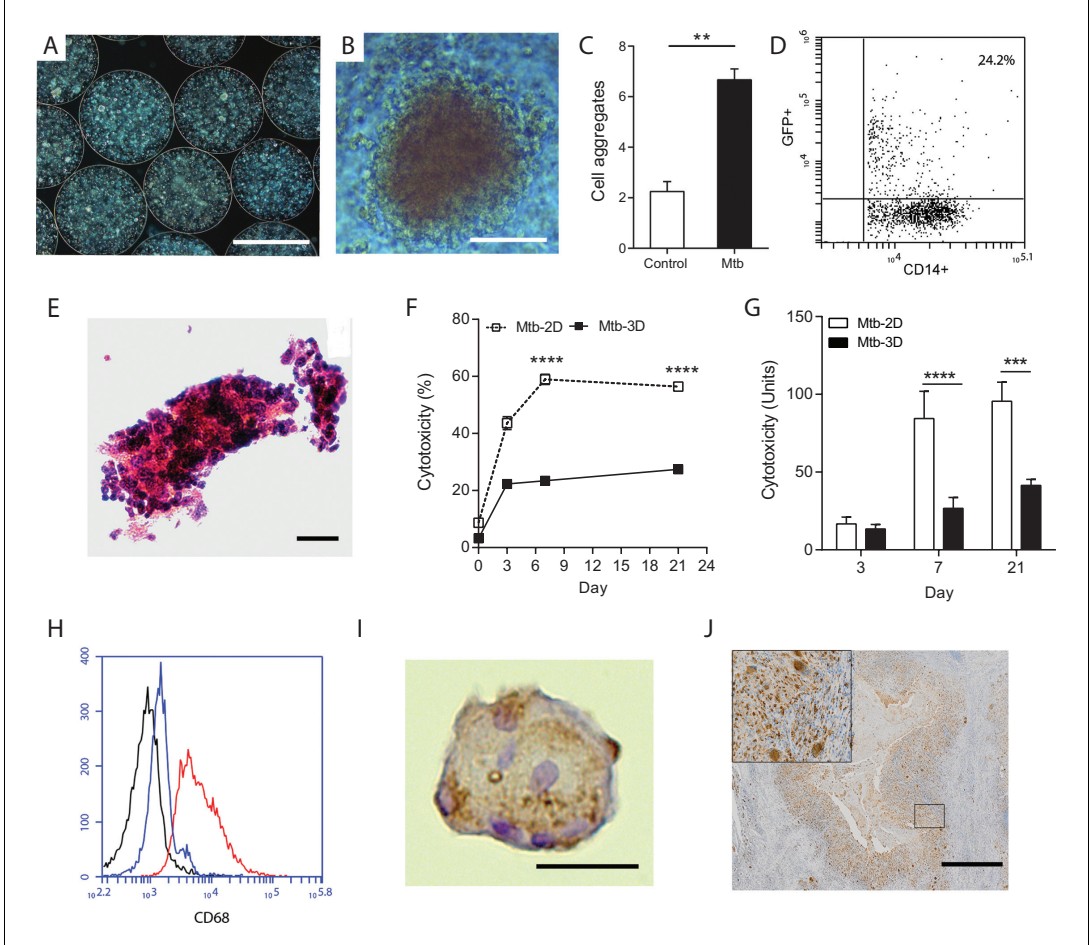

**Figure 1.** Primary human cells have greater survival in 3-D and aggregate, differentiate and fuse into multinucleate giant cells. (A) Phase contrast microscopy with overlay of Hoeschst 33256 (blue) at Day seven demonstrates PBMCs forming aggregates within microspheres. Scale 300 µm. (B) Cell aggregation in Mtb-infected PBMC-collagen-alginate microspheres at Day 7. Scale 50 µm. (C) Cell aggregation is greater in Mtb-infected microspheres than uninfected microspheres. Cells aggregates were defined as eight or more cells viewed under 20x magnification. Data are representative of a minimum of 10 fields of view per group. (D) Cells were infected with GFP+ Mtb and then released by decapsulation. At 24 hr after infection, 24.2% monocytes had phagocytosed GFP-expressing Mtb by flow cytometric analysis. (E) Haematoxylin and eosin staining of paraffin-fixed microspheres demonstrates cell aggregates in Mtb-infected microspheres at day 14. Scale 20 µm. (F) Host cell survival is significantly greater in 3-D microspheres than 2-D cell culture as demonstrated by LDH assay. Clear box 2-D cell culture, filled box 3-D culture; an equal number of cells killed with digitonin (30 µg/ml) in the respective 3D and 2D culture was used as denominator. Mean ± SE values (n = 4). (G) Cytotoxicity measured by CytoTox Glo assay is significantly lower in 3D culture than 2D culture. (H) CD68 expression is increased in macrophages in Mtb-infected microspheres analyzed by flow cytometry. Black isotype control, blue uninfected cells, red Mtb-infected cells. (I) Multinucleate giant cells form within microspheres at day 14, immunostained with CD68 (brown) and counterstained with Haematoxylin (blue). Scale 20 µm. (J) In patients with pulmonary TB, similar giant cells are observed in pulmonary granulomas. A low power image of human pulmonary granuloma (G), with numerous multinucleate giant cells surrounding caseous centre (box, magnified area). Scale bar: 1000 µm.

The following figure supplements are available for figure 1:

**Figure supplement 1.** Equipment set-up within containment level three tuberculosis laboratory.

**Figure supplement 2.** Microspheres placed in a 12 well tissue culture plate immediately after generation demonstrating non-magnified appearance.

**Figure supplement 3.** Medium viscosity guluronate (MVG) alginate is the optimal alginate for immunological studies.

**Figure supplement 4.** Viability of PBMCs remains over 90% after decapsulation.

**Figure supplement 5.** T cell composition of microspheres.

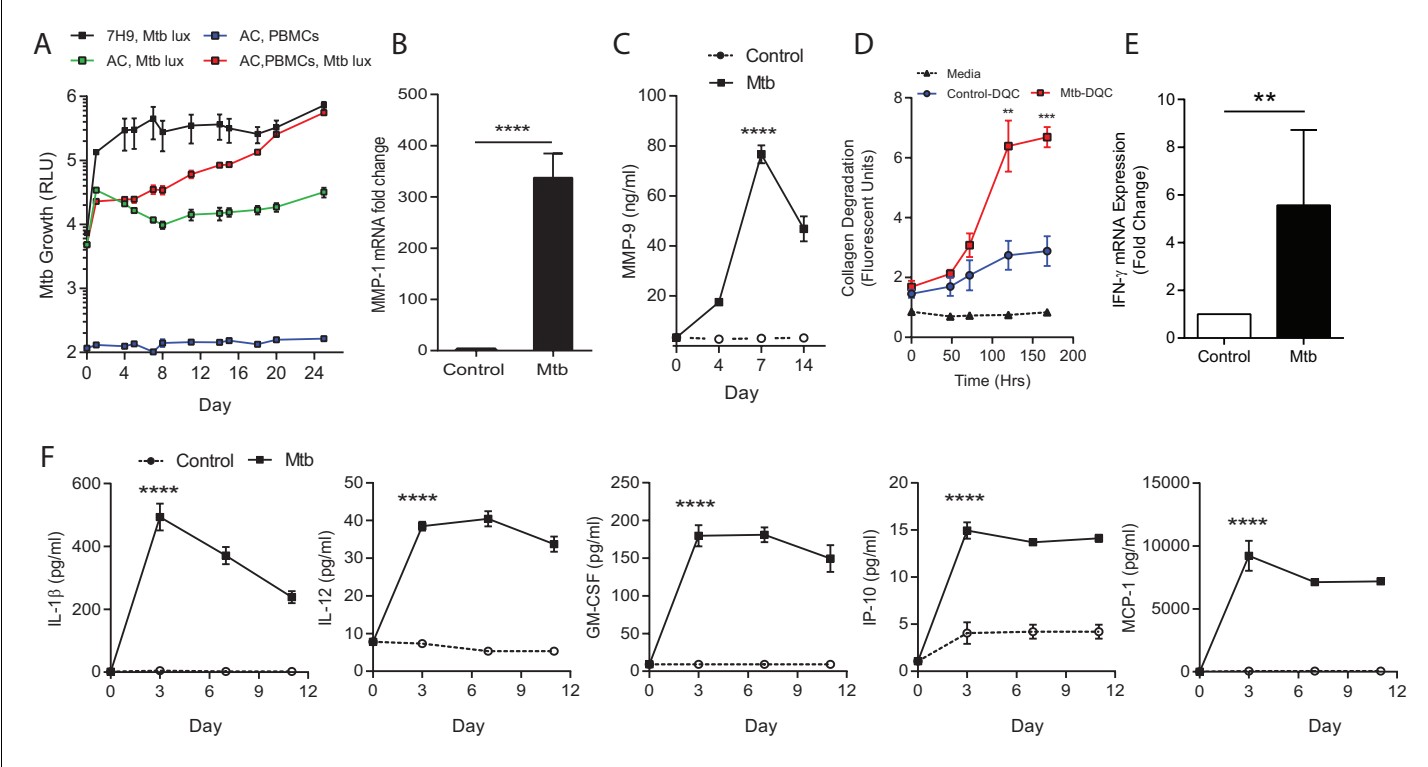

**Figure 2.** Mtb grows within microspheres containing PBMCs and upregulates MMP and cytokine expression. (A) Mtb proliferates slowly in microspheres with no cells (green line), but progressively in microspheres containing PBMCs (red line), reaching similar luminescence to Middlebrook 7H9 broth culture at 24 days (black line). Blue line, uninfected microspheres. (B) Mtb infection upregulates MMP-1 gene expression and (C) MMP-9 secretion in microspheres. (D) MMP upregulation has a functional effect, causing collagen degradation. DQ Collagen breakdown is higher in Mtb-infected microspheres (red line) than uninfected (blue line). Triangles, microspheres with no PBMCs. (E) Mtb infection increases cellular IFN-γ mRNA accumulation relative to uninfected cells at day four in microspheres (n = 4). (F) Secretion of cytokines by Mtb-infected microspheres (squares) is significantly higher than in microspheres containing uninfected PBMCs (circles). ****p<0.0001 by t-test (B and E) and ANOVA (A, C, D, F).

The following figure supplement is available for figure 2:

**Figure supplement 1.** Mtb infection upregulates secretion of multiple growth factors, cytokines and chemokines from microspheres measured by Luminex array.

## Defining the role of individual cytokines

An emerging paradigm within the TB field is that an optimal immune response is necessary, and that either a deficit or excess of specific mediators may be deleterious from the host's perspective (*O'Garra et al., 2013*). Therefore, we studied the effect of supplementing cytokines on both host and pathogen in collagen-containing microspheres, investigating IFN-γ and IFN-β. Complete absence of IFN-γ leads to disseminated Mtb infection in man and mouse (*O'Garra et al., 2013*), while IFN-β is of emerging importance from unbiased analyses but has an undefined mechanism of action (*Cliff et al., 2015*). Addition of exogenous IFN-β resulted in a minor but significant suppression of Mtb growth (*Figure 4A*), whereas in contrast addition of IFN-γ consistently increased growth (*Figure 4B*). We investigated mechanisms of this divergence. Both IFN-β and IFN-γ reduced cellular toxicity in Mtb-infected microspheres (*Figure 4C and D*). However, collagenase activity was divergently regulated, as IFN-β suppressed MMP-1 mRNA expression while IFN-γ increased MMP-1 expression (*Figure 4E*), suggesting that IFN-β has a matrix-protective role. Mtb infection upregulated IL-1β, TNF-α and MCP-1 secretion, but IFN augmentation did not significantly modulate this (*Figure 4F,G and H*). Mtb infection increased IFN-γ secretion, and this was further increased by IFN-β (*Figure 4I*), demonstrating complex cross-talk between these two cytokines.

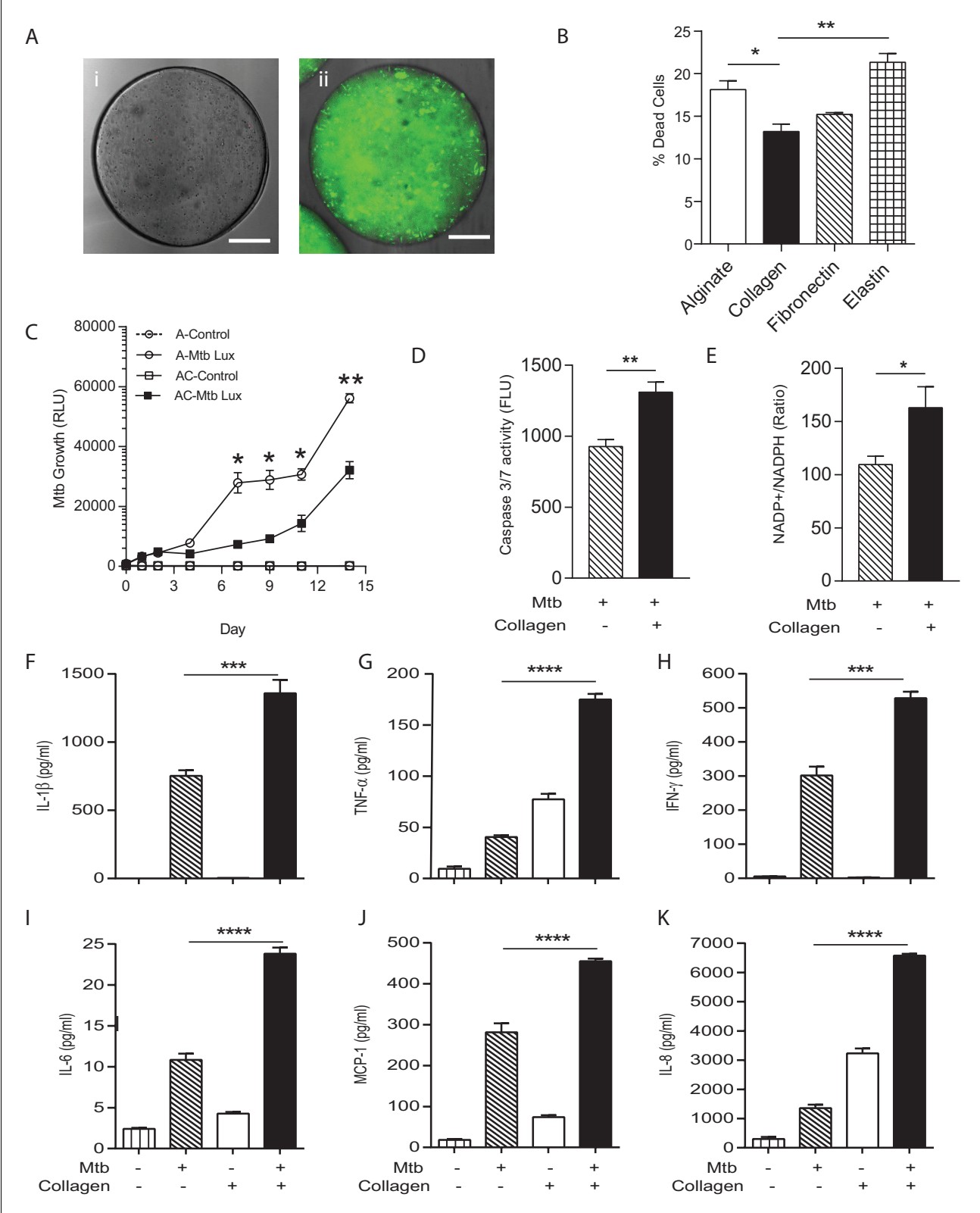

**Figure 3.** Incorporation of collagen into microspheres limits Mtb growth and increases host cell survival. (**A**) Microspheres were created without collagen (**i**), or incorporating FITC-labelled collagen (**ii**) to demonstrate distribution. Immediately after bioelectrospraying, collagen is homogenous throughout the microspheres. (**B**) Incorporation of Type I collagen into microspheres improves cell survival at 72 hr after Mtb infection, whereas elastin did not, analyzed by CytoTox-Glo assay. (**C**) PBMCs control Mtb growth in microspheres containing collagen (squares) better than cells without

*Figure 3 continued on next page*

*Figure 3 continued*

collagen (circles). Open squares, uninfected PBMCs. (D) The level of apoptosis and NADP+/NADPH ratio (E) are higher in microspheres containing collagen at day 7. Collagen incorporation caused increased secretion of IL-1$\beta$ (F), TNF-$\alpha$ (G), IFN-$\gamma$ (H), IL-6 (I), MCP-1 (J) and IL-8 (K) at day 7. Each experiment was performed a minimum of 2 times and charts represent mean values + SEM of a representative experiment performed in triplicate. *p<0.05, **p<0.01, ***p<0.001, ****p<0.0001.

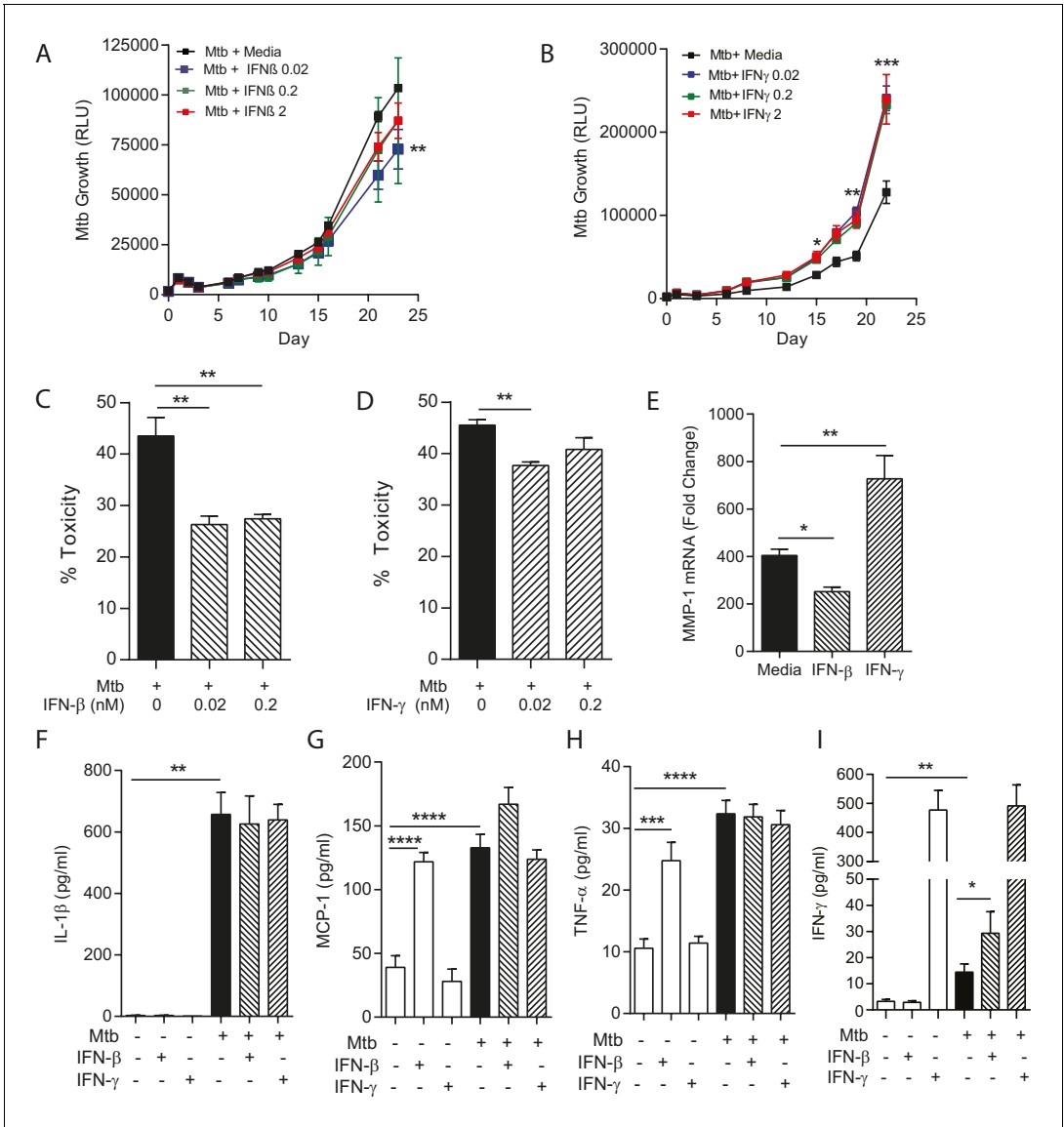

**Figure 4.** IFN-$\beta$ and IFN-$\gamma$ have divergent effects on bacterial growth within microspheres. (A) Exogenous IFN-$\beta$ suppresses Mtb growth after 24 days culture. Black line represents Mtb infected PBMCs. IFN-$\beta$ supplementation at 0.02 nM (blue), 0.2 nM (green) and 2 nM (red) suppresses Mtb luminescence. (B) IFN-$\gamma$ increases Mtb growth compared to infected PBMCs without additional cytokine. Exogenous IFN-$\gamma$ at 0.02 nM (blue), 0.2 nM (green) and 2 nM (red) increases Mtb luminescence above Mtb-infected PBMCs without cytokine supplementation (black line). (C, D) Both IFN-$\beta$ and IFN-$\gamma$ reduce toxicity in Mtb-infected PBMCs, analyzed by CytoTox-Glo assay. (E) IFNs divergently regulate MMP-1, with IFN-$\beta$ suppressing gene expression in infected microspheres while IFN-$\gamma$ increases MMP-1 expression. (F–H). Mtb upregulates cytokine secretion but this is not modulated by IFNs. IFN-$\beta$ drives TNF-$\alpha$ and MCP-1 as a single stimulus (E and G), but has no significant synergistic effect with Mtb. (I) Mtb upregulates IFN-$\gamma$ secretion, and this is further increased by the addition of IFN-$\beta$. Mean + SEM of a representative experiment performed in triplicate is shown, and are representative of a minimum of 2 experiments done in triplicate. *p<0.05, **p<0.01, ***p<0.001 and ****p<0.0001.

## Investigating host-directed therapy

Host-directed therapy is an emerging paradigm to improve outcome in TB infection (*Hawn et al., 2015*). However, the host immune response to Mtb has both beneficial and harmful effects, and so such therapy may inadvertently drive immunopathology whilst limiting mycobacterial proliferation (*Elkington and Friedland, 2015*). Modulation of the cyclooxygenase pathway has been proposed as a key target to limit Mtb growth (*Mayer-Barber et al., 2014*). Augmentation with exogenous $PGE_2$ suppressed Mtb growth in a dose-dependent manner (*Figure 5A*), consistent with findings in the mouse model of Mtb (*Mayer-Barber et al., 2014*). The reduced Mtb luminescence correlated with colony counts when microspheres were lysed and plated on Middlebrook 7H11 agar at day 22 (*Figure 5B*). However, this improved control of bacterial growth was not without potential harmful effects. Secretion of proinflammatory IL-6 and IL-8, a potent neutrophil chemoattractant, was increased by $PGE_2$ (*Figure 5C and D*). Conversely, secretion of IFN-γ was suppressed by high dose $PGE_2$ (*Figure 5E*). In addition, $PGE_2$ increased cell toxicity (*Figure 5F*) and suppressed total cell viability (*Figure 5G*). $PGE_2$ reduced caspase 3/7 activity, indicating suppression of apoptosis (*Figure 5H*). Therefore, the multiparameter readout has potential to predict protective and harmful effects of host-directed therapy.

## Immunoaugmentation with Mtb-specific T cell lines

The T cell response is critical to host control of Mtb but also drives pathology (*Kaufmann and Dorhoi, 2013*), and so a critical question is which facets of the adaptive immune response are protective versus those that are immunopathogenic (*Jasenosky et al., 2015*). We used the tractability of the bio-electrospray model to study T cell augmentation by supplementing PBMCs with autologous antigen specific T cell lines that had been proliferated ex vivo (*Figure 6—figure supplement 1*). Four days after bio-electrospraying, multicellular aggregates began to form containing PBMCs, Mtb and augmented T cells (*Figure 6A*). ESAT-6 or CFP-10 specific T cell lines, which respond to

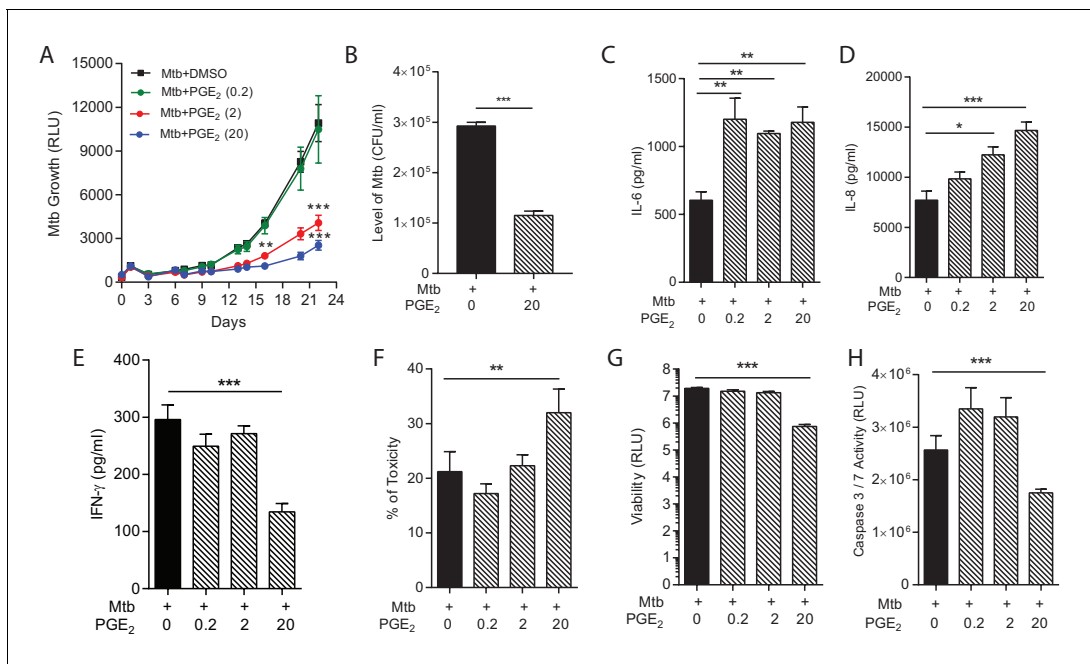

**Figure 5.** $PGE_2$ augmentation limits bacterial growth but increases pro-inflammatory cytokine secretion and cellular toxicity. (A) Addition of exogenous $PGE_2$ suppresses Mtb growth in microspheres in a dose-dependent manner. Mtb-infected PBMCs (black line), 0.2 µg/ml $PGE_2$ (green line), 2 µg/ml $PGE_2$ (red line), 20 µg/ml $PGE_2$ (blue line). (B) Colony counts of microspheres decapsulated at day 24 and then plated on Middlebrook 7H11 agar correlate with luminescence. (C, D and E) $PGE_2$ increases secretion of IL-6 and IL-8, but significantly decreases IFN-γ secretion, from Mtb-infected microspheres. (F) Cellular toxicity is increased in $PGE_2$ treated microspheres at day 3, analyzed by LDH release, and (G) total cell viability was reduced at day 7, analyzed by CytoTox-Glo assay. (H) $PGE_2$ reduces caspase 3/7 activity at day 7. *p<0.05, **p<0.01, ***p<0.001.

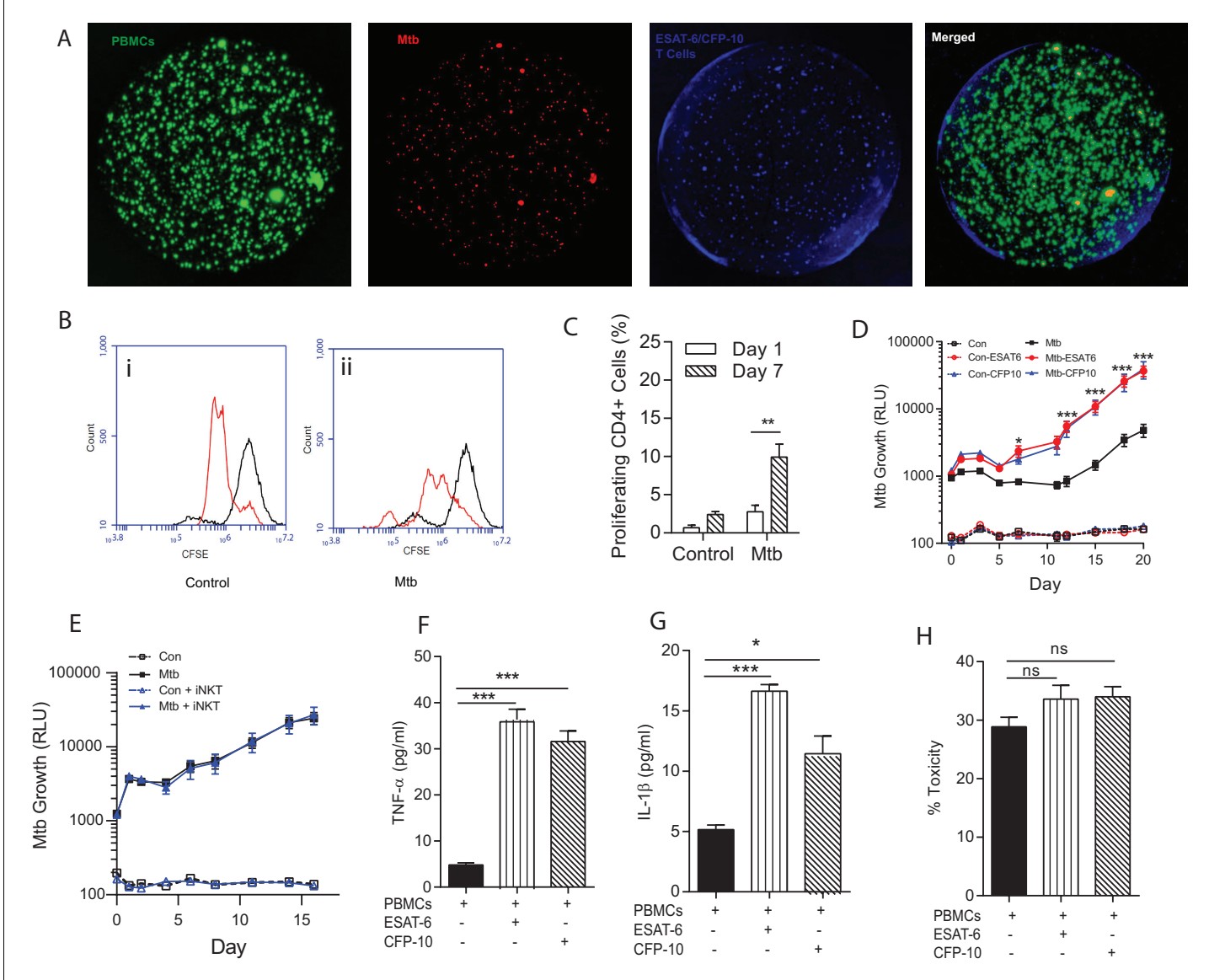

**Figure 6.** Immunoaugmentation with Mtb-specific T cells increases Mtb growth. (A) Microspheres imaged after 4 days show early granuloma formation. (i) PBMCs labelled with CellTrace CFSE (green), (ii) Mtb expressing mCherry (red), (iii) autologous ESAT-6 specific T cells labelled with CellTracker Blue, (iv) Merged image shows granulomas containing Mtb, PBMCs and augmented T cells (yellow). (B) Cellular proliferation is increased in infected microspheres with immunoaugmented autologous T cells, analysed by CFSE staining. Day 1, black line; Day 7, red line; (i) Uninfected, (ii) Mtb-infected. (C) Quantitative analysis of the proliferative capacity of ESAT-6 augumented PBMCs at Day one and Day 7. The bars show percentage of proliferating CD4 cells after gating on CD3+CD4+ lymphocytes. Differences between Day 1 and 7 were assessed for three experiments by t-test. (D) Addition of either ESAT-6 responsive T cells (red) or CFP-10 responsive T cells (blue) increases Mtb growth compared to infected PBMCs without supplemented T cells (black). Open symbols, uninfected microspheres. (E) Supplementation with an iNKT autologous T cell line (blue triangle) did not significantly affect Mtb growth compared to infected PBMCs alone (black square). (F, G) Secretion of TNF-α and IL-1β is increased in immunoaugmented microspheres at day 7. (H) Immunoaugmentation did not significantly modulate cell toxicity in infected microspheres at day three analysed by LDH release.

The following figure supplements are available for figure 6:

**Figure supplement 1.** Confirmation of specificity of in vitro expanded T cells.

**Figure supplement 2.** Augmentation of PBMCs with ESAT-6/CFP-10 specific T cell lines in microspheres causes differential secretion of cytokines after Mtb infection compared to PBMCs alone.

antigens secreted via the pathogenicity RD1 locus, proliferated in the Mtb infected microspheres but not uninfected spheres (*Figure 6B*), which was statistically significant from day seven on quantitation (*Figure 6C*). Surprisingly, immunoaugmentation with either ESAT-6 or CFP-10 T cell lines led to increased Mtb growth within microspheres compared to infected PBMCs alone (*Figure 6D*). Augmentation with an autologous innate iNKT cell line has no effect on Mtb growth (*Figure 6E*), demonstrating that this was not a generic response to T cell supplementation within the microspheres. Immunoaugmentation with ESAT-6 and CFP-10 lines significantly increased secretion of multiple cytokines, including TNF-α and IL-1β into the media around microspheres (*Figure 6F and G* and *Figure 6—figure supplement 2*). In contrast, augmentation did not significantly affect cell toxicity within infected microspheres (*Figure 6H*).

## Discussion

Novel approaches to the global TB pandemic are urgently required. Mtb is an obligate human pathogen characterised by a prolonged interaction with the host (*Russell, 2011*; *Cambier et al., 2014*), a spatially organised immune response (*Marakalala et al., 2016*; *Mattila et al., 2013*; *Egen et al., 2008*) and extensive extracellular matrix turnover (*Elkington et al., 2011*). Therefore, extended studies of human cells adherent to extracellular matrix in 3-D are likely to be essential to fully understand the host-pathogen interaction. We developed a model of human TB utilising bio-electrospray technology that replicates key features of clinical disease and optimised a multiplex readout to investigate both host and pathogen responses. We demonstrated that the extracellular matrix regulates the host immune response, consistent with reports of the ECM regulating inflammation (*Sorokin, 2010*), and found that collagen favours host control of Mtb. We then used the model to investigate novel therapeutic approaches. The system permitted prolonged culture of primary human cells, and we found that significant differences between experimental conditions often only emerged after more than 7 days, which would not have been observed in 2D culture systems where 3–4 days is the standard experimental duration. Our findings are consistent with other infections, where the cellular adaptions to their context determines outcome (*Snijder et al., 2009*). This cell culture platform is highly flexible for both matrix and cellular composition within spheres and therefore has wide potential applicability within the biomedical field.

Our bioengineering approach differs significantly from traditional model systems to investigate TB, which predominantly rely on culture of human cells in 2D culture systems without extracellular matrix, infection of the zebrafish larvae with *M. marinum* or infection of mice with Mtb (*Young, 2009*; *Guirado and Schlesinger, 2013*; *Vogt and Nathan, 2011*). The mouse model of TB has many advantages and key findings in the mouse have been replicated in man, such as critical roles for CD4 + T cells, TNF-α and IFN-γ (*Flynn and Chan, 2001*), but pathology in the mouse differs from human TB (*Young, 2009*) and humanised mice are required to generate caseating lesions (*Calderon et al., 2013*; *Heuts et al., 2013*). Other advanced human cellular models of TB have been developed. For example, Altare's group has studied a prolonged model of PBMC culture with Mtb and demonstrated cell aggregate formation (*Puissegur et al., 2004*; *Lay et al., 2007*), but this model lacks extracellular matrix. A collagen matrix-containing model has been developed by Kapoor, showing aggregation and TB dormancy (*Kapoor et al., 2013*), but lacks the high throughput potential of the bioelectrospray system and rapid cellular recovery for multiparameter readouts. Generation of a complete human granuloma structure will require stromal cells such as fibroblasts, which are present in the periphery of TB granulomas (*O'Kane et al., 2010*). A key next step will be to compare patients with latent TB with active TB, and those with and without HIV-co-infection, to determine whether the model can differentiate protective immunity to Mtb directly ex vivo.

Many of our findings are consistent with conclusions drawn from these systems. For example, a significant role for IFN-β in the host immune response to TB is emerging from genomic studies, though it remains controversial as to whether this is protective or harmful (*Cliff et al., 2015*). Our data suggest a predominantly protective effect, and emerging data support this conclusion (*Moreira-Teixeira et al., 2016*). Similarly, we confirmed that augmentation of PGE$_2$ improves host control of mycobacterial proliferation, consistent with findings in the mouse (*Mayer-Barber et al., 2014*). Finally, T cells responsive to specific Mtb antigens proliferated in infected microspheres and secreted cytokines known to be important in the host immune response to Mtb (*O'Garra et al., 2013*).

However, while some results were as expected, several findings in the 3-D system may not be predicted from current disease paradigms. For example, IFN-γ in high concentrations increases the growth of TB, whereas murine experiments predict improved control. Consistent with our findings, several previous studies have shown that IFN-γ increases Mtb growth in primary human cells (*Vogt and Nathan, 2011*; *Douvas et al., 1985*; *Rook et al., 1986*; *Crowle and Elkins, 1990*). The evidence for a beneficial role of IFN-γ in humans is principally supported by individuals where there is a complete absence of signalling through the IL-12/IFN pathway (*Karp et al., 2015*), and this protective effect has clearly been shown in the mouse through both knock-out and vaccination studies (*Flynn and Chan, 2001*; *Aagaard et al., 2011*). However, cohort studies have shown that a high PPD response, or high IFN-γ response to ESAT-6 or CFP-10, associates with the subsequent development of TB (*Comstock et al., 1974*; *Higuchi et al., 2008*; *del Corral et al., 2009*), suggesting that an excessive IFN-γ response may be deleterious. We attempted to determine if there was a tipping point of IFN-γ concentration by adding IFN-γ neutralising antibodies to the microsphere matrix, and although we were able to demonstrate increased growth with IFN-γ neutralisation, we found a similar effect with isotype control antibodies, so it was impossible to determine if this was a specific effect. Our longitudinal observations of Mtb growth in a non-destructive manner using luminescent mycobacteria support the emerging concept that a balanced immune response is essential, and either a deficit or excess of a specific mediator may favour the pathogen (*O'Garra et al., 2013*; *Gideon et al., 2015*). Combining the model with CRISPR/Cas9 gene editing will permit further interrogation of each cytokine component of the immune response.

PGE$_2$ augmentation has been proposed as a novel host-directed therapy to improve outcome in TB, and we demonstrate reduced Mtb growth. However, the multiparamter analysis in our human system showed that PGE$_2$ increases secretion of IL-8, which is likely to drive migration of neutrophils, and PGE$_2$ also reduced host cell viability. Neutrophil recruitment has been described to have a deleterious effect on host control of infection (*Kimmey et al., 2015*; *Nouailles et al., 2014*) and therefore there is potential that this may favour Mtb, driving increased pathology, lung destruction and transmission. The peak PGE$_2$ concentration that we studied is similar to that reported in human tissue (*Reikerås et al., 2009*). We also found that immunoaugmentation with ESAT-6 or CFP-10 responsive T cell lines led to increased growth of Mtb. Pathogenic mycobacteria express the RD1 locus but the precise mechanism linking RD1 to pathology is not fully understood (*Majlessi et al., 2015*). Our findings are consistent with the recent observations that T cell epitopes are hyperconserved in pathogenic mycobacteria (*Comas et al., 2010*; *Coscolla et al., 2015*; *Lindestam Arlehamn et al., 2015*), indicating an evolutionary advantage to the pathogen of specifically stimulating components of the host immune response to facilitate transmission (*Orme et al., 2015*). However, an alternative explanation is that the in vitro expansion conditions generated a T cell phenotype that was permissive to Mtb growth, skewing an initially protective phenotype to a deleterious one, and so further confirmation across different T cell lines is required. Augmented iNKT cells did not increase Mtb growth after ex vivo expansion. The key conclusion of these experiments is that the immunoaugmentation model has potential to dissect protective versus pathological host immune responses.

The recent negative outcomes from both vaccine trials (*Tameris et al., 2013*; *Ndiaye et al., 2015*) and treatment-shortening regimes (*Warner and Mizrahi, 2014*) illustrate that observations in current model systems may not reliably translate to human disease and highlight the need for more nuanced approaches that reflect human TB infection. Our data suggest that simply driving an increased immune response to Mtb will not improve control of mycobacterial growth. Augmenting PGE$_2$ release by modulating the leukotriene pathway may reduce mycobacterial proliferation but may come at a cost of increased pathology. Vaccination using ESAT-6 as an antigen, which has entered human trials (*Luabeya et al., 2015*), may actually favour Mtb growth under certain circumstances, demonstrating the fine balance between the host immune response and control of pathogen growth. Critically, our data from the bioengineered model are consistent with clinical phenomena observed in human TB. For example, an excessive immune response in patients is associated with greater pulmonary pathology (*Kaufmann and Dorhoi, 2013*; *Comstock et al., 1974*; *Philips and Ernst, 2012*; *Nunes-Alves et al., 2014*). Similarly, our immunoaugmentation studies concur with the expression of ESAT-6 and CFP-10 by pathogenic mycobacteria, implying a critical role in causing disease (*Brites and Gagneux, 2015*). The model can be used to investigate approaches currently in development, such as vaccines based on targeting CD1-restricted T cells (*Van Rhijn*

*et al., 2015*) and emerging host-directed therapies (*Hawn et al., 2013*; *Zumla et al., 2015*), to determine whether they confer greater protection without likely harmful effects.

The bio-electrospray cell culture model has broad potential, addressing the technical complexity of performing 3-D primary cell culture within diverse extracellular matrices. The system can readily be applied to study diverse infectious and inflammatory diseases, or cancer immunotherapy, and can be developed for high-throughput applications by combining the microsphere system with microfluidics. Integration with CRISPR/Cas9 gene editing will permit genetic manipulation of both host and pathogen (*Chakraborty et al., 2014*). The multiparameter readouts define the translational potential of novel interventions over time with longitudinal data acquisition, identifying both beneficial and deleterious effects. Therefore, this system developed to dissect the host-pathogen interaction in human TB can be applied to identify novel therapeutic approaches to multiple human diseases.

## Materials and methods

### Ethical approval
For analysis of blood from healthy donors and healthy TB exposed individuals, this work was approved by the National Research Ethics Service committee South Central - Southampton A, study title 'An investigation into the immune response to tuberculosis infection and development of novel diagnostic markers', reference 13/SC/0043. All donors gave written informed consent. For histological analysis, samples used in this study were sourced from the Southampton Research Biorepository, University Hospital Southampton NHS Foundation Trust and University of Southampton, Mailpoint 218, Tremona Road, Southampton, SO16 6YD. Lung biopsy tissue was taken as part of routine clinical care and tissue blocks excess to diagnostic testing were analyzed in this study. The project was approved by the Institutional Review Board (Reference 12/NW/0794 SRB04_14). The ethics committee approved the analysis of this tissue without individual informed consent since it was surplus archived tissue taken as part of routine care.

### PBMC cell isolation from human blood
PBMCs were isolated from single donor leukocyte cones (National Health Service Blood and Transfusion, Southampton, UK) or fresh blood from volunteers by density gradient centrifugation over Ficoll-Paque (GE Healthcare Life Sciences). These healthy donors were all recruited from a region of very low TB incidence. For immunoaugmentation experiments requiring *Mycobacterium tuberculosis*-responsive T cells, cells from donors with a documented tuberculosis exposure were studied. All experiments were performed with primary human cells; no immortalised cell lines were used in the study.

### *M. tuberculosis* culture
*M. tuberculosis* H37Rv (Mtb) was cultured in Middlebrook 7 H9 medium (supplemented with 10% ADC, 0.2% glycerol and 0.02% Tween 80) (BD Biosciences, Oxford) and bioluminescent *M. tuberculosis* H37Rv (*Andreu et al., 2010*), GFP or mCherry expressing *M. tuberculosis* H37Rv (*Carroll et al., 2010*) were cultured with kanamycin 25 µg/ml and hygromycin 50 µg/ml, respectively. Bioluminescent Mtb H37Rv was used for all experiments apart from confocal imaging. Cultures at $1 \times 10^8$ CFU/ml Mtb (OD = 0.6) was used for all experiments at multiplicity of infection (M.O.I) of 0.1. For colony counting, Mtb was released from microspheres by EDTA/sodium citrate dissolution, cells and extracellular bacteria were pelleted by centrifugation at 3000 g, lysed with 1% saponin and then plated on Middlebrook 7H11 agar. Colonies were counted at three weeks.

### Cell encapsulation
Microspheres were generated with an electrostatic generator (Nisco, Zurich, Switzerland) as described previously (*Workman et al., 2014*). To visualise microsphere formation, a Phantom v7 high-speed camera, capable of capturing 150000 fps in conjunction with a long-distance microscope lens, was triggered simultaneously with a fibre optic lighting system to capture the jetting process. For cellular encapsulation, PBMCs were infected with Mtb overnight in a T75 flask. Cells were then detached, washed and pelleted by centrifugation at 320 g and mixed with 1.5% sterile alginate (Pronova UP MVG alginate, Nova Matrix, Norway) or alginate with human collagen, fibronectin (both

from Advanced BioMatrix) or human elastin (Sigma Aldrich, Gillingham, UK) at a final concentration of $5 \times 10^6$ cells/ml. The standard matrix used in experiments was alginate-human collagen unless stated otherwise. All reagents were confirmed endotoxin free.

The protocol for the encapsulation of PBMCs in alginate based matrix is described in detail at Bio-protocol (*Tezera et al., 2017*). The cell-alginate suspension was drawn up into a sterile syringe and injected into the bead generator at 10 mL/hour via a Harvard syringe driver through a 0.7 mm external diameter nozzle. Microspheres of 600 µm diameter formed in an ionotropic gelling bath of 100 mM CaCl$_2$ in HBSS placed below the electrostatic ring that accelerated the microspheres from the needle head. After washing twice with HBSS with Ca/Mg, microspheres were transferred in RPMI 1640 medium containing 10% of human AB serum and incubated at 37°C, 21% O$_2$ and 5% CO$_2$. No media changes were performed, and the supernatant was harvested at defined time points for analysis. Mtb growth within microspheres was monitored longitudinally by luminescence (GloMax 20/20 Luminometer, Promega).

## Immunofluorescence and confocal imaging

In specific imaging experiments, PBMCs were pre- labelled with CellTracker Blue, CellTrace CFSE or Hoechst 33342 (ThermoFisher Scientific, UK) according to the manufacturer recommendation. Microspheres were fixed in 4% paraformaldehyde and washed in HBSS with Ca/Mg. Confocal images were acquired on a Leica TCS SP5 Confocal microscope and processed using Image J 1.5 0d (NIH, USA).

## Histological staining and immunohistochemistry

On day 7 and 14 of incubation, microspheres were fixed with 4% paraformaldehyde overnight and paraffin-embedded using the Shandon Cytoblock system (ThermoFisher Scientific, UK). Blocks were sectioned and stained by haematoxylin and eosin. For CD68 immunohistochemistry (Dako, Clone PG-M1), 0.5 µm sections were stained. Analysis of human lung tissue taken as part of routine clinical care was approved by the Institutional Review Board (Reference 12/NW/0794 SRB04_14). Sections were dewaxed, blocked (Envision FLEX), stained with Anti-Human CD68 (Dako, Clone PG-M1), detected with HRP and counterstained with Haematoxylin.

## Flow cytometric analysis

Cells were extracted by dissolving the microspheres in 55 mM Sodium Citrate and 10 mM EDTA in PBS for 10 min at 37°C. Cells were then suspended in PBS. Surface and intracellular staining was done in a three-colour analysis with combinations of fluorescein isothiocyanate (FITC), phycoerythrin (PE) and allophycocyanin (APC). Antibodies used included anti-CD3, anti-CD4, anti-CD8, anti-CD14 and anti-CD68 (ImmunoTools, Germany). For T–cell proliferation, PBMCs were stained with Cell-Trace CFSE Cell Proliferation Kit for flow cytometry (ThermoFisher Scientific, UK) before infection with Mtb. Fluorescence was then analyzed by flow cytometry (BD Accuri C6 flow cytometer).

## MMP-1 and IFN-γ gene expression

Microspheres were decapsulated using 55 mM of sodium citrate solution and cells were lysed immediately using TRIzol Reagent (Life Technologies, Paisley, UK). 1 µg RNA was reverse transcribed using High Capacity cDNA Reverse Transcription kit (Life Technologies Ltd, Paisley, UK). Taqman Universal master mix and primers specific for GAPDH, β-Actin, MMP1, and IFN-γ gene were used for qPCR following manufacturer's instruction (Applied Biosystems, USA) and comparative threshold (CT) method was employed to analyse all qPCR data.

## Luminex analysis

Samples were sterilized by filtration through a 0.22 µM Durapore membrane (Millipore) (*Elkington et al., 2006*). Concentrations of cytokines (Life Technologies, UK) and MMPs (R and D Systems, UK) were determined using a Bioplex 200 platform (Bio-Rad, UK) according to the manufacturer's protocol.

## DQ collagen degradation assay

For analysis of extracellular matrix degradation, microspheres were generated from a solution of 3% Alginate (Pronova UP MVG alginate, Nova Matrix, Norway), 1 mg/ml of human collagen type I (Advanced BioMatrix, San Diego, California) and 100 µg/ml of DQ collagen (Invitrogen, Paisley, UK). Microspheres were then placed in macrophage serum free medium (Life Technologies) and incubated at 37°C. Fluorescence was read on a GloMax Discover (Promega) at an absorption maxima of 495 nm and fluorescence emission maxima of 515 nm.

## Cell viability assay

Microspheres were incubated in 96-well plates at 37°C. Cell viability was analyzed using the Cell-Titer-Glo 3D Cell Viability Assay (Promega) according to the manufacturer's instructions, analyzing viable cells based on ATP quantitation. Luminescence was analyzed by the GloMax Discover plate reader (Promega).

## Cell toxicity assays

CytoTox-Glo Cytotoxicity Assay (Promega) measured cellular necrosis in microspheres. The kit measures dead-cell protease activity released from cells without membrane integrity using a luminogenic peptide substrate, the AAF-Glo Substrate. Luminescence was analyzed by GloMax Discover (Promega). Total cell death was caused by digitonin on equal cell numbers to provide the denominator. As a second measurement of toxicity, lactate dehydrogenase (LDH) release, was analyzed by a colorimetric activity assay (Roche, Burgess Hill, United Kingdom). Comparison of 2D cell culture and 3D cell culture viability was performed by plating equal numbers of cells in wells of a 96 well plate, and then measuring toxicity by LDH and CytoTox-Glo. Total toxicity was normalised to digitonin treated wells plated in parallel.

## Cell apoptosis

Microspheres were incubated in 96-well plates. Caspase-3/7 protease activities were measured as indicators of apoptosis using Apo-ONE Homogeneous Caspase-3/7 Assay (Promega) or Caspase-Glo 3/7 Assay Systems (Promega) according to the manufacturer's instructions.

## NADP/NADPH ratio

The biolumiesent NADP/NADPH kit (Promega) was used according to the manufacturer's instructions.

## Cytokine and PG supplementation

Microspheres were incubated in RPMI 1640 with 10% AB serum with IFN-$\beta$, IFN-$\gamma$ (eBioscience) or PGE$_2$ (R and D Systems) at 37°C.

## Immunoaugmentation with autologous T cells

To generate expanded specific T cell lines, ESAT-6 or CFP-10 specific cells were derived from PBMCs from Mtb-exposed individuals (*Wölfl and Greenberg, 2014*). Monocytes were isolated from PBMCs using magnetic cell separation (Miltenyi Biotec, UK) and partially differentiated into monocyte derived dendritic cells (moDCs) for 3 days using complete media supplemented with GM-CSF (20 ng/ml) and IL-4 (25 ng/ml). moDCs were then loaded with peptide antigen pools derived from ESAT-6 or CFP-10 proteins in the presence of IFN-$\gamma$ and LPS. moDCs were then exposed to CD14$^-$ T cell fractions in a 1:2 ratio for 7 days, after which IL-2 (400IU/ml, Proleukin, Chiron), IL-15 (2 ng/ml) and IL-7 (2 ng/ml, Immunotools) were added. T-cell specificity was confirmed by IFN-$\gamma$ secretion upon exposure of T-cells to autologous moDCs loaded with CFP-10 or ESAT-6. Human iNK T cells were derived from PBMCs according to the method previously described (*Mansour et al., 2015*). Briefly, iNK T cell lines were generated by incubating PBMCs with 200 ng/ml KRN7000 (AXXORA) for 7 days before the addition of IL-2, IL-7 and IL-15. After two weeks culture, iNK T cell expansion was confirmed via CD3, Va24 and CD1d-K7 tetramer staining. Cells were acquired using FACSAria (Becton Dickinson, UK). ESAT-6, CFP-10 specific T cell lines or iNK T cells were counted and then PBMCs were supplemented with 20% additional immunoaugmented cells immediately prior to Mtb

infection. After overnight incubation, the combined cells were bioelectrosprayed by our standard protocol.

## Statistical analysis

All experiments were performed a minimum of 2 occasions from separate donors as biological replicates and on each occasion with a minimum of 3 technical replicates. Data presented are from a representative donor and include the mean and SEM. Analysis was performed in Graphpad Prism v6.0. Students t-test was used to compare pairs and ANOVA for groups of 3 or more.

## Acknowledgements

This work was supported by the US National Institute for Health R33AI102239, the UK National Centre for the 3Rs NC/L001039/1 and the Antimicrobial Resistance Cross Council Initiative supported by the seven research councils MR/N006631/1. We would like to thank Jennifer Russell and Regina Teo, University of Southampton, for excellent technical assistance. We thank Nuria Andreu and Siouxsie Wiles for providing the Lux-expressing Mtb, Brian Robertson for the GFP+ Mtb and Tanya Parish for the mCherry expressing Mtb. The authors declare no conflict of interest.

## Additional information

### Funding

| Funder | Grant reference number | Author |
|---|---|---|
| Medical Research Council | MR/N006631/1 | Paul T Elkington |
| National Institute of Allergy and Infectious Diseases | R33AI102239 | Suwan N Jayasinghe Paul T Elkington |
| National Centre for the Replacement, Refinement and Reduction of Animals in Research | NC/L001039/1 | Liku B Tezera Paul T Elkington |

The funders had no role in study design, data collection and interpretation, or the decision to submit the work for publication.

### Author contributions

LBT, Conceptualization, Data curation, Formal analysis, Investigation, Methodology, Writing—original draft; MKB, Data curation, Formal analysis, Investigation, Writing—review and editing; AC, Resources, Formal analysis, Investigation, Writing—review and editing; MTR, BAS, PB, Investigation, Writing—review and editing; AB, Investigation, Performed experiments as part of rotational project; AT, Resources, Investigation, Writing—review and editing; SJ, Resources, Methodology, Writing—review and editing; BGM, Resources, Project administration, Writing—review and editing; MT, Resources, Investigation, Project administration, Writing—review and editing; SNJ, Conceptualization, Resources, Methodology, Writing—review and editing; SM, Conceptualization, Investigation, Methodology, Writing—review and editing; PTE, Conceptualization, Data curation, Formal analysis, Supervision, Funding acquisition, Investigation, Methodology, Writing—original draft, Project administration, Writing—review and editing

### Author ORCIDs

Paul T Elkington, http://orcid.org/0000-0003-0390-0613

### Ethics

Human subjects: For analysis of blood from healthy donors and healthy TB exposed individuals, this work was approved by the National Research Ethics Service committee South Central - Southampton A, study title "An investigation into the immune response to tuberculosis infection and development of novel diagnostic markers", reference 13/SC/0043. All donors gave written informed consent. For histological analysis, samples used in this study were sourced from the Southampton Research Biorepository, University Hospital Southampton NHS Foundation Trust and University of Southampton, Mailpoint 218, Tremona Road, Southampton, SO16 6YD. Lung biopsy tissue was taken as part of

routine clinical care and tissue blocks excess to diagnostic testing were analyzed in this study. The project was approved by the Institutional Review Board (Reference 12/NW/0794 SRB04_14). The ethics committee approved the analysis of this tissue without individual informed consent since it was surplus archived tissue taken as part of routine care.

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
