## [Decision Letter]

Thank you for submitting your article "Dissection of the host-pathogen interaction in human tuberculosis using a bioengineered 3-dimensional model" for consideration by *eLife*. Your article has been reviewed by Bree Aldridge (Reviewer #1) and JoAnne Flynn (Reviewer #2), and the evaluation has been overseen by a Reviewing Editor and Richard Losick as the Senior Editor.

The reviewers have discussed the reviews with one another and the Reviewing Editor has drafted this decision to help you prepare a revised submission.

Summary:

Tezera et al. describe the use of a biospray system to create beads in which to study *M. Tuberculosis*-host interactions in a 3D environment. There is a particular need in TB research to establish new model systems. A simple 3D bead-based system that more closely mimics the 3D environment of the human host may accelerate our understanding of TB biology and the development of improved interventions. The reviewers believe that the 3-D biospray model for human tuberculosis described here may be an important tool for the field, but that the strength of the model must be further illustrated with a few additional experiments. We invite a revised manuscript that describes these experiments and addresses the concerns listed below:

Essential revisions:

1) Although mouse models can be wrong, the importance of T cells specific for Mtb antigens, including ESAT-6 or CFP-10, has been clearly demonstrated. Models where ESAT-6 or CFP-10 T cells provide protection alone have been published. Therefore, the exacerbation of growth when specific T cells were added to the microsphere must be further investigated and validated. First, a titration of T cells should be performed – is there a tipping point for numbers of T cells, and how was the number of T cells determined? In addition, what is the phenotype of those T cells in terms of cytokine production (a feature of how they were generated)? Since this result is not intuitive, it must be further investigated.

2) It would also be very helpful for a comparison of microspheres using PBMC from Mtb+ (active or latent TB) humans to microspheres with PBMC from uninfected individuals. This is a much more realistic and interesting comparison.

3) To validate the IFN-γ results, the authors could consider antibodies against IFN-γ to determine if the added IFN-γ pushes the system over the top. Can the authors measure IFN-γ in their system – this would be important here and in the T cell experiments.

4) In the Abstract, please list the cell types used in the study.

5) The proliferation data in the T cell study shows only modest proliferation. This should be quantified over several different microspheres and provided in graphical form, in addition to the histograms provided.

6) Figure 1 does not look like a human granuloma. In Figure 1, it is simply an aggregate of cells, not an organized granuloma. This statement should be clarified and the absence of true structure should be noted. If they have evidence of true granuloma structure, they should provide it. Otherwise, this is very misleading.

7) The PGE_2_ experiments are very interesting and provocative. How does the 20μg/ml concentration use relate to in vivo concentrations in murine models?

8) The methodology is difficult to understand or is not fully described: Novel methods such as the spray technology have a very brief discussion in the main text and should be given more space given that this is a tools article. It was difficult, for example, to understand what cells were used and that they were preinfected. Information on how the cells were infected was not included, for example, nor was how the bacteria were cultured.

Other aspects of the methods that are emphasized as part of the general utility of the model are not well described. Only by reading the Methods do I see that cytokines are measured using xMAP technology. It is not clear without reading Andreu et al., 2010 how the luminescence assay is longitudinal.

How is decapsulation performed? Given how important decapsulation is to the functionality of this model for studying the host-pathogen interaction, it is important to demonstrate how efficient and damaging the decapsulation process is.

How were the CFU experiments performed? After how many days were the cells plated?

It is hard to understand how comparable the 2D and 3D cytotoxicity measurements are (Figure 1) without information about how many cells were used in the measurements.

9) The emphasis in the writing is on characterizing bacterial burden and host cell behaviors, but falls short of comprehensively placing these findings in context of experiments performed in animal models and in 2D.

---

## [Author Response]

*Summary:*

*Tezera et al. describe the use of a biospray system to create beads in which to study M. Tuberculosis-host interactions in a 3D environment. There is a particular need in TB research to establish new model systems. A simple 3D bead-based system that more closely mimics the 3D environment of the human host may accelerate our understanding of TB biology and the development of improved interventions. The reviewers believe that the 3-D biospray model for human tuberculosis described here may be an important tool for the field, but that the strength of the model must be further illustrated with a few additional experiments. We invite a revised manuscript that describes these experiments and addresses the concerns listed below:*

We thank the reviewers for their comments and for seeing the potential of the model for the field. We have done as many experiments as technically possible within the 2 month requested period for resubmission. However, due to the prolonged time course of the experiments, which each often last for 21 days, it has not been possible to do all experiments suggested, and we have amended the text where appropriate to respond to the specific comments and highlight future direction for the model.

*Essential revisions:*

*1) Although mouse models can be wrong, the importance of T cells specific for Mtb antigens, including ESAT-6 or CFP-10, has been clearly demonstrated. Models where ESAT-6 or CFP-10 T cells provide protection alone have been published. Therefore, the exacerbation of growth when specific T cells were added to the microsphere must be further investigated and validated. First, a titration of T cells should be performed – is there a tipping point for numbers of T cells, and how was the number of T cells determined? In addition, what is the phenotype of those T cells in terms of cytokine production (a feature of how they were generated)? Since this result is not intuitive, it must be further investigated.*

We agree with the points the reviewers make. We have performed detailed characterisation of the cytokine response of the ESAT-6 and CFP-10 augmented T cells and present these data as new Figure 6—figure supplement 1, and Figure 6—figure supplement 2 – E. There is significantly increased secretion of IL-6, IL-8, TNF-α, IL-1β, MCP-1 and IFN-γ secretion in augmented microspheres. We have amended the Discussion (fourth paragraph) to include the evidence that ESAT-6 can be protective, and also cite human data where an increased IFN-γ response correlates with the subsequent development of TB. These contrasting observations demonstrate that the host immune response to TB is highly nuanced and a greater response does not consistently correlate with greater protection.

In terms of T cell titration, although we completely agree that they are very valuable suggestions, the experiments suggested are simply not possible in the time frame available, as expansion of the T cells would take 20 days, the microsphere incubation would be 20 days, and then colony counts 21 days, so just one experiment would take 2 months. We accept that the ex vivo proliferation may have created an adverse T cell phenotype, and so have amended the text discussing whether the expansion protocol has led to a non-controlling phenotype (Discussion, fifth paragraph). We have also included future potential experiments to investigate this possibility.

Critically, the fundamental conclusion that the model can be used to distinguish protective versus pathological T cell responses is demonstrated by our data and supports the use of the model to investigate the host-pathogen interaction.

*2) It would also be very helpful for a comparison of microspheres using PBMC from Mtb+ (active or latent TB) humans to microspheres with PBMC from uninfected individuals. This is a much more realistic and interesting comparison.*

We completely agree that this is a valuable line of investigation and indeed is a programme of work we would like to undertake. However, allowing for inter-donor variability that is inevitable in a primary human system, we anticipate needing to study at least 6 uninfected and 6 latently infected individuals, again at least 4 months’ work. Therefore, it is simply impossible in the time frame available. We now discuss these experiments in the second paragraph of the Discussion and hope to be able to undertake this work in the coming year.

*3) To validate the IFN-γ results, the authors could consider antibodies against IFN-γ to determine if the added IFN-γ pushes the system over the top. Can the authors measure IFN-γ in their system – this would be important here and in the T cell experiments.*

We thank the reviewers for this suggestion. We have measured IFN-γ in the system and also analysed gene expression by RT-qPCR. Mtb infection upregulates gene expression and secretion. We now include these data as Figure 2 and Figure 4. In the cytokine augmentation experiment, we show that Mtb infection upregulates IFN-γ secretion, and this is further increased by IFN-β (Figure 4). In the PGE_2_ experiment, we show that PGE_2_ suppresses IFN-γ secretion, and include this as Figure 5. In the immunoaugmentation experiments, there was a significant increase in IFN-γ secretion and this is presented in Figure 6—figure supplement 2.

In terms of adding antibodies, we have performed this experiment as requested, and again shown in 3 further donors that IFN-γ increases TB growth, so we have now demonstrated this result in a total of 8 donors studied. We found that anti-IFN-γ antibodies also increased TB growth, consistent with the tipping point hypothesis proposed by the reviewers. However, we also found a similar effect with the isotype control antibody, so impossible to know if this is a specific effect. We now discuss these results in the fourth paragraph of the Discussion, and make it clear that further work, such as by CRISPR/Cas9 gene editing, is required to resolve this this.

*4) In the Abstract, please list the cell types used in the study.*

The cell types are now stated.

*5) The proliferation data in the T cell study shows only modest proliferation. This should be quantified over several different microspheres and provided in graphical form, in addition to the histograms provided.*

Thank you, we have performed quantitation as suggested from 3 separate donors and provide this as Figure 6. Statistical analysis shows a significant increase in proliferation after T cell augmentation at day 7.

*6) Figure 1 does not look like a human granuloma. In Figure 1, it is simply an aggregate of cells, not an organized granuloma. This statement should be clarified and the absence of true structure should be noted. If they have evidence of true granuloma structure, they should provide it. Otherwise, this is very misleading.*

We accept the point made and that this is a cell aggregation and further work is required to prove it is an organised granuloma. We have changed the wording throughout the manuscript from granuloma to aggregate. We now discuss the future potential development of the model in the second paragraph of the Discussion to generate a fully multicellular system.

*7) The PGE_2_ experiments are very interesting and provocative. How does the 20μg/ml concentration use relate to* in vivo *concentrations in murine models?*

We are glad the reviewers find the experiments interesting, and highlight the potentially double-edged nature of TB immunotherapy. The concentration we used was chosen on the basis of those reported in human inflamed lung, which is 20μg/ml, and we now cite this reference in the manuscript. We have been unable to find reports of local concentrations of PGE_2_ in the mouse model, and justify the concentration studied on the basis of human data.

*8) The methodology is difficult to understand or is not fully described.*

We apologise for this and accept that our familiarity with the system may have led us to describe it poorly. We have amended throughout to give greater detail, addressed the points below.

*Novel methods such as the spray technology have a very brief discussion in the main text and should be given more space given that this is a tools article. It was difficult, for example, to understand what cells were used and that they were preinfected. Information on how the cells were infected was not included, for example, nor was how the bacteria were cultured.*

We have expanded this section in the Results section giving an overview of the process and we have provided additional details in the Methods. Most importantly, we have initiated the submission of a Bio-protocol giving step by step guides of the entire process, and we will complete the submission immediately if this article is accepted for publication in *eLife* to make it freely available to researchers worldwide.

*Other aspects of the methods that are emphasized as part of the general utility of the model are not well described. Only by reading the Methods do I see that cytokines are measured using xMAP technology. It is not clear without reading Andreu et al., 2010 how the luminescence assay is longitudinal.*

We have clarified these two points in the Results in the last paragraph of the subsection “Key features of human tuberculosis develop in the bio-electrospray model” and re-read the manuscript to try to ensure that all methods are described as clearly as possible.

*How is decapsulation performed? Given how important decapsulation is to the functionality of this model for studying the host-pathogen interaction, it is important to demonstrate how efficient and damaging the decapsulation process is.*

We give further details on decapsulation in the second paragraph of the subsection “Key features of human tuberculosis develop in the bio-electrospray model”. We thank the reviewers for this comment and now include data from 3 experiments studying cell toxicity of decapsulation and include this as Figure 1—figure supplement 4. Cells have over 92% viability after the entire matrix incorporation / encapsulation / decapsulation process compared to autologous cells stored at 4^o^C during this process. Therefore, we feel that the process is minimally damaging.

*How were the CFU experiments performed? After how many days were the cells plated?*

We apologise for omitting these data. CFUs were performed by decapsulating cells, lysing with 1% saponin and the plating on Middlebrook 7H11 agar. Plates were read at 3 weeks. We stipulate the time point taken more clearly in the Results and Methods sections.

*It is hard to understand how comparable the 2D and 3D cytotoxicity measurements are (Figure 1) without information about how many cells were used in the measurements.*

We now provide greater details about the cell numbers used and how the measurements were performed in the revised Figure 1 legend and in the Methods subsection “DQ Collagen Degradation Assay”.

*9) The emphasis in the writing is on characterizing bacterial burden and host cell behaviors, but falls short of comprehensively placing these findings in context of experiments performed in animal models and in 2D.*

We accept this criticism and now include an entirely new paragraph in the Discussion trying to place the model more comprehensively in findings from animal and 2D systems.